# Long-term effectiveness and safety of infliximab-biosimilar: A multicenter Phoenix retrospective cohort study

Tomoe Kazama[1], Katsuyoshi Ando[2], Nobuhiro Ueno[2], Mikihiro Fujiya[2], Takahiro Ito[3], Atsuo Maemoto[3], Keisuke Ishigami[1], Masanori Nojima[4], Hiroshi Nakase[1]*

1 Department of Gastroenterology and Hepatology, Sapporo Medical University School of Medicine, Sapporo, Japan, 2 Division of Metabolism and Biosystemic Science, Gastroenterology, and Hematology/Oncology, Department of Medicine, Asahikawa Medical University, Asahikawa, Japan, 3 Sapporo Higashi Tokushukai Hospital IBD Center, Sapporo, Japan, 4 Center for Translational Research, The Institute of Medical Science Hospital, The University of Tokyo, Tokyo, Japan

* hiropynakase@gmail.com

## Abstract

### Background

Infliximab (IFX) effectively treats patients with inflammatory bowel disease (IBD). IFX-biosimilar (IFX-BS) has the same amino acid sequence as that of the IFX originator, and its increasing use is expected to reduce national healthcare costs. Long-term efficacy and safety of IFX-BS in patients with Crohn's disease (CD) and ulcerative colitis (UC) have not been completely investigated.

### Methods

We conducted a retrospective, multicenter observational study of patients with IBD who received IFX-BS treatment at three hospitals between October 2016 and April 2022. Clinical data were collected from electronic medical records and evaluated for achieving clinical remission (CR) using Crohn's disease activity index (CDAI) and partial Mayo (pMayo) score, persistency of long-term IFX-BS administration, and clinical response rate in the bio-naïve and bio-failure groups.

### Results

A total of 117 patients with IBD (90 CD and 27 UC) were included. The study findings indicated that both bio-naïve and bio-failure groups of patients with UC showed similar effectiveness of IFX-BS. The treatment persistence rate in patients with CD was significantly higher in the bio-naïve (P = 0.042) and switch (P = 0.010) groups than in the bio-failure group. In the former two groups, the treatment persistence rate was high at two years after administration (more than 80%). In patients with UC, the findings indicated higher treatment persistence rate in the switch group than in the bio-naïve group. Univariable and multivariable analyses for treatment persistence rate showed that the albumin level at the initial IFX-BS

**Data Availability Statement:** All relevant data are within the manuscript.

**Funding:** This work was supported by the Health and Labour Sciences Research Grants for research

on intractable diseases from the Ministry of Health, Labor and Welfare (MHLW) of Japan (Investigation and Research for intractable Inflammatory Bowel Disease) (Grant Number 20FC1037).

**Competing interests:** H Nakase has received support from AbbVie, Celgene, Daiichi Sankyo, EA Pharma, Janssen, JIMRO, Kissei Pharmaceutical, Kyorin Pharmaceutical, Mitsubishi Tanabe Pharma, Mochida Pharmaceutical, Nippon Kayaku, Pfizer, Takeda, and Zeria Pharmaceutical, as well as grants for commissioned/joint research from Boehringer Ingelheim, Bristol Myers Squibb, and Pentax Medical. Dr. Fujiya acknowledges grants from Japanese Grants-in-Aid for Scientific Research (21K07929), grants from Translational Research Network Program of Japan Agency for Medical Research and Development (B-118), non-financial support from Development and Intractable Disease Health and Labour Sciences Research Grants from the Ministry of Health, Labour and Welfare. Dr. Fujiya reports grants, personal fees from EA Pharma Co., Ltd., grants, personal fees from AYUMI Pharmaceutical Corporation, grants, personal fees from AbbVie Inc, grants, personal fees from Otsuka Pharmaceutical Co., Ltd., grants, personal fees from ZERIA Pharmaceutical Co., Ltd., grants, personal fees from Nippon Kayaku Co., Ltd., grants, personal fees from Nobelpharma Co., Ltd., grants, personal fees from Pfizer Inc, grants, personal fees from Janssen Pharmaceutical K.K., grants, personal fees from KYORIN Pharmaceutical Co., Ltd., grants, personal fees from MOCHIDA PHARMACEUTICAL CO.,LTD., grants, personal fees from Daiichi Sankyo Company, Limited, grants, personal fees from Mitsubishi Tanabe Pharma Corporation, grants, personal fees from Takeda Pharmaceutical Company Limited, grants, personal fees from Yakult Honsha Co., Ltd., personal fees from OLYNPUS Co., Ltd., personal fees from celltrionhealthcare.jp, personal fees from Alfresa Pharma Corporation, personal fees from Mylan Inc., personal fees from Boston Scientific Corporation, personal fees from Covidien Japan, Inc., personal fees and non-financial support from FUJIFILM Corporation, grants from Fuji Chemical Industries Co., Ltd., grants from JIMRO Co., Ltd., grants from Kamui Pharma. Inc. K Ando has received lecture fees from Nippon Kayaku Co. Ltd., Janssen Pharmaceutical K.K., Mitsubishi Tanabe Pharma Corporation, AYUMI Pharmaceutical Corporation, Pfizer Inc., Takeda Pharmaceutical Co. Ltd., AbbVie GK., JIMRO Co. Ltd., EA Pharma Co. Ltd., Mochida Pharmaceutical Co. Ltd., Kyorin Pharmaceutical Co. Ltd., ZERIA Pharmaceutical Co. Ltd., Aspen Japan K.K., Sandoz K.K. and research grant from Pfizer Inc. N Ueno has received

administration and groups (bio-naïve, bio-failure and switch) were effective factors for patients with CD. Adverse events were reported in 18 patients (15.4%).

## Conclusion

The present study demonstrates the long-term effectiveness and safety of IFX-BS. In addition to the favorable remission induction in the bio-naïve and bio-failure groups, we demonstrated remission maintenance and treatment persistence rates beyond two years. Albumin level and groups were associated with better treatment persistence in patients with CD.

## Introduction

Inflammatory bowel disease (IBD), including Crohn's disease (CD) and ulcerative colitis (UC), is characterized by relapsing and remitting mucosal inflammation. Tumor necrosis factor-α (TNF-α), an inflammatory cytokine, is closely involved in the pathogenesis of IBD. Anti-TNF-α agents such as infliximab (IFX, commercialized at May 2002, Mitsubishi Tanabe Pharma Corporation, Osaka, Japan), are highly effective in the induction and maintenance of remission in IBD and have remarkably improved patients' quality of life [1, 2]. On the contrary, they are expensive and have a significant impact on total national healthcare costs. IFX-biosimilars (IFX-BSs) developed in Korea were cheaper than first-generation anti-TNF-α agents. They have been approved for use in the treatment of rheumatoid arthritis (RA), ankylosing spondylitis, CD, UC, and psoriasis. The use of IFX-BS has gradually increased since the Japanese government's approval in 2014. Economic benefits of using IFX-BS in RA patients have also been reported [3]. Several clinical trials have demonstrated the efficacy and safety of IFX-BS in the treatment of autoimmune diseases [4–7]. In addition, real-world data have been reported from South Korea [8], Japan [9], the United Kingdom [10], Norway [11], Hungary [12], and several other countries, but the long-term effectiveness and safety of IFX-BS have not been sufficiently investigated.

In this study, we investigated the long-term effectiveness and safety profile of IFX-BS and identified the factors associated with treatment persistence rate in patients with IBD treated with IFX-BS. Our study is based on data from the cohort of patients with IBD (Principal research in Hokkaido Organization Emphasizing Nutritional and therapeutic Improvement to IBD patients' eXpectation cohort; Phoenix cohort).

## Materials and methods

### Study design and participants

We conducted a multicenter, retrospective observational study in Hokkaido, Japan, including patients from three hospitals, namely Sapporo Medical University Hospital, Asahikawa Medical University Hospital, and Sapporo Higashi Tokushukai Hospital. We enrolled patients with CD and UC who received IFX-BS (Infliximab Biosimilar 1, commercialized on Nov 2014, Nippon Kayaku Co., Ltd. Tokyo, Japan and Infliximab Biosimilar 3, commercialized on Dec 2018, Pfizer Inc., NY, USA) in the period from October 2016 to April 2022. All patients were diagnosed with CD or UC based on specified criteria. CD is a chronic inflammatory disease of unknown etiology characterized by noncontiguous distributed, all-stratified granulomatous inflammation and fistulae. UC is a diffuse, nonspecific inflammation of unknown origin that continuously damages the colonic mucosa from the rectal side, often leading to erosions and ulcers [13].

personal fees from AbbVie Inc, AYUMI
Pharmaceutical Corporation, EA Pharma Co. Ltd.,
Janssen Pharmaceutical K.K., JIMRO Co. Ltd.,
Kyorin Pharmaceutical Co. Ltd., Mitsubishi Tanabe
Pharma Corporation, Mochida Pharmaceutical Co.
Ltd., Nippon Kayaku Co. Ltd., and Takeda
Pharmaceutical Co. Ltd. A Maemoto and T Ito have
received support from Takeda Pharmaceutical
Company, Eli Lilly Japan K.K., Janssen
Pharmaceutical K.K., Gilead Sciences, Inc., Nippon
Boehringer Ingelheim Co., Ltd., AbbVie GK, Pfizer
R&D Japan G.K., EA Pharma Co., Ltd., Kaken
Pharmaceutical Co., Ltd., Mochida Pharmaceutical
Co., Ltd., Kissei Pharmaceutical Co., Ltd.,
Mitsubishi Tanabe Pharma Corporation. This does
not alter our adherence to PLOS ONE policies on
sharing data and materials.

In this study, we categorized the patients with IBD into three groups: bio-naïve, bio-failure, and switch from originator to IFX-BS (switch) groups. The bio-naïve group included patients who never received biologics and Janus kinase (JAK) inhibitors, the bio-failure group included patients who had insufficient prior treatment such as IFX originator, vedolizumab, adalimumab, and JAK inhibitors, and the switch group included patients who received IFX originator in remission at baseline and switched to IFX-BS. During the study period, data were evaluated at 8, 30, and 54 weeks after administration of IFX-BS for short-term data, and at 2 (104 weeks), 3 (156 weeks), 4 (208 weeks), and 5 years (260 weeks) for long-term data. The clinical characteristics were collected from electronic medical records.

## Ethics statements

This study was approved by the Institutional Review Boards of Sapporo Medical University School of Medicine (IRB No. 302–101) and other participating medical centers. We provided information about this study on the hospital website and gave participants the opportunity to opt-out of the study; we considered patients who did not opt-out to be providing tacit consent for participation.

## Treatment

In the bio-naïve and bio-failure groups, the dose of IFX-BS at induction was 5 mg/kg of body weight at 0, 2, and 6 weeks, and every 8 weeks thereafter. In the switch group, the dose of IFX-BS continued to be the same as that in the originator. In all groups of patients with CD, the dose of IFX-BS was increased to 10 mg/kg, and the treatment interval was shortened to 4 weeks, depending on their symptoms.

## Outcomes

In patients with CD, clinical remission (CR) was defined as Crohn's disease activity index (CDAI) < 150 points, and clinical response was defined as a decrease in at least 70 points of CDAI from baseline. In patients with UC, CR was defined as a pMayo score of two points or fewer, and clinical response was defined as a decrease in the pMayo score of at least 30% from baseline. Endpoints of this study were the proportion of patients achieving CR in the switch group and treatment persistence of long-term administration of IFX-BS. We also evaluated the clinical response in the bio-naïve and bio-failure groups, and long-term safety during the observational period. To assess the factors contributing to the treatment persistence rate in patients with CD, the patients were analyzed for sex, age, groups including bio-naïve group, bio-failure group, and switch group, hemoglobin level, albumin level, C-reactive protein (CRP) level, concomitant use of immunomodulators as explanatory variables. For the safety profile of IFX-BSs, all patients were asked about adverse events, including infusion reactions, infection, hospitalization, and any potential adverse events at every visit. Patients who were lost to follow-up or had missed clinical scores were considered censored. When the CDAI or pMayo could not be calculated because of missing data, we excluded them from subsequent analyses for effectiveness. Due to the frequent missing of follow-up, we additionally applied the inverse probability of censoring weighting (IPCW) method to estimate the probability of the clinical remission rate in CD patients for each time point. The probability of censoring was estimated using baseline measurements (sex, clinical disease type, prior medications, dose escalation of prior anti-TNF-α inhibitor medication, shortened duration of prior anti-TNF-α inhibitor therapy, nutritional therapy, concomitant use of 5-ASA, steroids, and immunomodulators, albumin level, hemoglobin level, CRP level, CDAI score, history of intestinal resection, IFX-BS dose escalation, shortened IFX-BS duration) using a logistic regression model. We

then estimated the remission rate by weighting the inverse probability of uncensored until each time point using generalized estimation equation with robust variance estimates.

## Statistical analysis

The covariates associated with treatment persistence rates were assessed by univariable and multivariable analyses using Cox proportional hazards regression analysis. The results are expressed as hazard ratio (HR) and 95% confidence intervals (95% CIs). Cox proportional hazards regression was performed with EZR v1.61 (Saitama Medical Center, Jichi Medical University, Saitama, Japan), which is a graphical user interface for R (The R Foundation for Statistical Computing, Vienna, Austria) [14]. Treatment persistence was assessed by the Kaplan–Meier method and log-rank test using JMP® Pro (version 16.2.0, SAS Institute, Cary, NC). P values <0.05 were considered statistically significant. All authors had access to the study data and reviewed and approved the final manuscript.

## Results

### Patient characteristics

A total of 117 patients with IBD (90 CD and 27 UC) were included. The baseline characteristics of the included patients are shown in Tables 1–3. Patients with CD were 17 in the bio-naïve group, three in the bio-failure group, and 70 in the switch group. L3 (ileocolon) was the most common disease location of CD (71.1%). The median CDAI score at baseline for bio-naïve, bio-failure, and switch groups in patients with CD was 192, 259, and 53.3, respectively. Patients with UC were 14 in the bio-naïve group, seven in the bio-failure group, and six in the switch group. In patients with UC, E3 (pancolitis) was the most common disease extent (20/ 27, 74.1%). The median pMayo score at baseline for bio-naïve, bio-failure, and switch groups was 6, 7, and 0, respectively. Two of the three patients with CD who showed loss of response (LOR) to prior biologics were administered with IFX originators, and one was administered adalimumab. On the contrary, three of the ten patients with UC who showed LOR to prior biologics and JAK inhibitor were administered adalimumab, four vedolizumab, and three tofacitinib.

### Clinical effectiveness

In CD patients, the percentages of responders in the bio-naïve and bio-failure groups at 8 weeks were 81.8% (9/11) and 0% (0/3), respectively (Fig 1A). In the switch group, CR rates at 8 weeks, 54 weeks, 2 years, 3 years, and 5 years were 97% (64/66), 87.2% (34/39), 72.0% (18/25),

**Table 1. Base line characteristics of the enrolled patients.**

|  | CD (n = 90) | UC (n = 27) |
|---|---|---|
| Sex (male, n (%)) | 66 (73.3) | 17 (63.0) |
| Age (y.o, Median (IQR)) | 45 (37–53.5) | 49 (29–65) |
| Onset of disease (y.o, Median (IQR)) | 25 (19–31) | 34 (20–53) |
| Alcohol (n (%)) | 14 (15.6) | 7 (25.9) |
| Smoking (n (%)) | Current 7 (7.8); Past 5 (5.6) | Current 2 (7.4); Past 8 (29.6) |
| Year of inclusion | 2/ 30/ 12/ 20/ 14/ 10/ 2 | 1/ 7/ 4/ 10/ 4/ 1/ 0 |
| (2016/2017/2018/2019/2020/2021/2022) |  |  |
| Observation periods (weeks, Median(IQR)) | 120 (54–169) | 71.1 (7–133) |

IQR, inter quartile range

**Table 2. Clinical characteristics of the CD patients.**

|  | CD (n = 90) | | |
|---|---|---|---|
| **Disease status** | | | |
| Disease location[a] (L1/L2/L3/L4, n (%)) | 17 (18.9) / 9 (10.0) / 64 (71.1) / 0 (0) | | |
| History of intestinal resection (n (%)) | 39 (43.3) | | |
| Anal lesion (n (%)) | 45 (50.0) | | |
| Extra intestinal lesion[b] (n (%)) | 9 (10.0) | | |
| **Group** | **Naïve[c]** | **Failure[c]** | **Switch[c]** |
| Patients (n (%)) | 17 (18.9) | 3 (3.3) | 70 (77.8) |
| Observation periods (weeks, Median(IQR)) | 81 (49–120) | 6 (4.1–77) | 129 (56–178) |
| (Mean) | 56.4 | 63.7 | 69.0 |
| (Person years) | 0.080 | 0.65 | 0.12 |
| Baseline CDAI (Median (IQR)) | 192 (90–262) | 259 (203–283) | 53.3 (31.4–90.6) |
| **Laboratory data at baseline** | | | |
| Hb (g/dL) (Median (IQR)) | 12.1 (11.2–13) | 10.2 (9.4–12) | 13.7 (12.8–14.8) |
| Albumin (g/dL) (Median (IQR)) | 3.7 (3.3–3.8) | 3.3 (2.9–3.8) | 4.1 (3.9–4.4) |
| CRP (mg/dL) (Median (IQR)) | 0.69 (0.34–2.3) | 1.6 (0.80–3.6) | 0.10 (0.10–0.12) |
| **Concomitant medication** | | | |
| 5-ASA (n (%)) | 71 (78.9) | | |
| Steroid (n %)) | 3 (3.3) | | |
| IM (n (%)) | 37 (41.1) | | |
| **Prior biologics or JAK inhibitor[d]** | | | |
|  | Failure | | Switch |
| IFX originator | 2 | | 70 |
| Adalimumab | 1 | | 0 |
| Vedolizumab | 0 | | 0 |
| Tofacitinib | 0 | | 0 |

IQR, inter quartile range; 5-ASA, 5-Aminosalicylic acid; IM, Immunomodulator.

[a]Disease location in CD patients was assessed according to the Montreal classification at the time of initial administration of IFX-BS. L1 = terminal ileum, L2 = colon, L3 = ileocolon, L4 = upper gastro-intestinal.

[b]Extra intestinal lesion includes arthritis, erythema nodosum, primary biliary cholangitis, and ocular complications in CD patients.

[c]Naïve, Failure, and Switch indicate bio-naïve, bio-failure, and switch groups, respectively.

[d]Includes some duplicates

61.9% (13/21), and 60.0% (12/20), respectively (Fig 1B). In the bio-naïve group, CRP and CDAI at 8 weeks after IFX-BS administration was significantly lower than baseline, and mean CDAI remained less than 150 during the follow-up period (S1, S2 Appendices). In the bio-failure group, all three patients who showed LOR to the previous treatment with anti-TNF-α agents (IFX:2, adalimumab:1) did not respond to IFX-BS administration. In the switch group, the median CDAI remained less than 100 and CRP did not vary as significantly as in the other groups; however, large deviation cases were observed each week.

The results of the clinical remission rates in the switch group with CD patients corrected by the IPCW method are shown in S1 Table. CR rates at 30 weeks, 54 weeks, 2 years, 3 years, 4 years, and 5 years were 0.90 (95%CI: 0.76–0.96), 0.85 (0.62–0.95), 0.78 (0.52–0.92), 0.74 (0.44–0.91), 0.78 (0.50–0.93), and 0.77 (0.49–0.92), respectively (S1 Table). Clinical remission rates were improved in comparison to those before the IPCW analysis.

**Table 3. Clinical characteristics of the UC patients.**

| | UC (n = 27) | | |
|---|---|---|---|
| **Disease status** | | | |
| Disease extent[a] (E1/E2/E3, n (%)) | 3 (11.1) / 4 (14.8) / 20 (74.1) | | |
| History of intestinal resection (n (%)) | 0 (0) | | |
| Extra intestinal lesion[b] (n (%)) | 2 (7.4) | | |
| **Group** | **Naïve[c]** | **Failure[c]** | **Switch[c]** |
| Patients (n (%)) | 14 (51.9) | 7 (25.9) | 6 (22.2) |
| Observation periods (weeks, Median(IQR)) | 76 (16–136) | 76 (16–136) | 76 (16–136) |
| (Mean) | 62.0 | 62.0 | 62.0 |
| (Person years) | 0.32 | 0.32 | 0.32 |
| Baseline pMayo score (Median (IQR)) | 6 (5–7) | 7 (6.5–7.5) | 0 (0–0.8) |
| **Laboratory data at baseline** | | | |
| Hb (g/dL) (Median (IQR)) | 11.2 (10.4–12.8) | 11.6 (9.8–12.4) | 14.4 (14.3–14.7) |
| Albumin (g/dL) (Median (IQR)) | 3.4 (3.1–3.9) | 3.1 (2.6–3.2) | 4.2 (4.1–4.3) |
| CRP (mg/dL) (Median (IQR)) | 0.89 (0.23–2.0) | 1.2 (0.51–2.0) | 0.10 (0.10) |
| **Concomitant medication** | | | |
| 5-ASA (n (%)) | 20 (87.0) | | |
| Steroid (n %)) | 5 (21.7) | | |
| IM (n (%)) | 15 (65.2) | | |
| **Prior biologics or JAK inhibitor[d]** | | | |
| | Failure | | Switch |
| IFX originator | 0 | | 6 |
| Adalimumab | 3 | | 0 |
| Vedolizumab | 4 | | 0 |
| Tofacitinib | 3 | | 0 |

IQR, inter quartile range; 5-ASA, 5-Aminosalicylic acid; IM, Immunomodulator.

[a]Disease extent in UC patients was assessed according to the Montreal classification at the time of initial administration of IFX-BS. E1 = proctitis, E2 = left-sided colitis, E3 = pancolitis.

[b]Extra intestinal lesion includes arthritis and thrombosis in UC patients.

[c]Naïve, Failure, and Switch indicate bio-naïve, bio-failure, and switch groups, respectively.

[d]Includes some duplicates

In patients with UC, the percentages of responders in the bio-naïve and bio-failure groups at 8 weeks were 63.6% (7/11) and 57.1% (4/7), respectively (Fig 2A). Even at 54 weeks, the responder rates were 60% (6/10) and 42.9% (3/7), respectively, indicating treatment effectiveness in both groups. In the switch group, CR rates at 8 weeks, 54 weeks, 2 years, 3 years, and 5 years were 100% (6/6), 75.0% (3/4), 33.3% (1/3), 66.7% (2/3), and 50% (1/2), respectively (Fig 2B). The pMayo score and CRP for the bio-naïve and bio-failure groups showed improvement over time (S3, S4 Appendices).

## Treatment persistence rate

In patients with CD, treatment persistence rates at two years were 85.9% (95%CI: 57.4–96.5) in the bio-naïve group, and 86.0% (75.0–92.6) in the switch group, respectively. In both groups, the treatment persistence rate two years after administration was more than 80%, indicating a high treatment persistence rate, which was significantly higher in the bio-naïve (P = 0.042) and switch (P = 0.010) groups than in the bio-failure group (Fig 3). In patients with UC, the

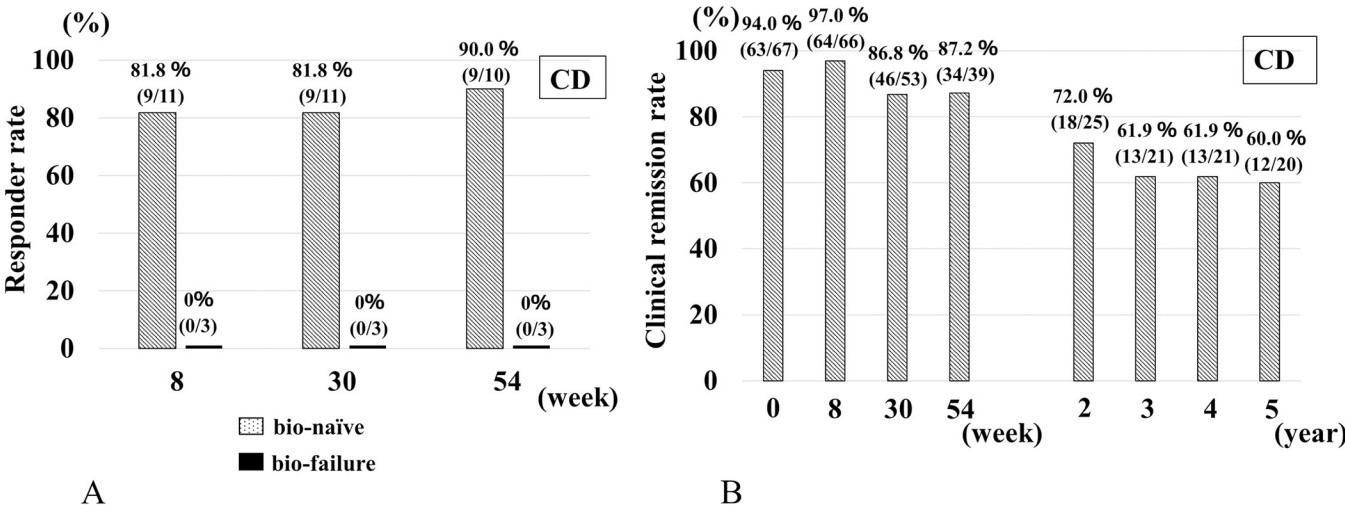

**Fig 1. Effectiveness of IFX-BS in patients with CD.** (A) Responder rate in patients with CD at baseline, 8, 30, and 54 weeks. (B) Clinical remission rate in patients with CD at baseline, 8, 30, and 54 weeks, and 2, 3, 4, and 5 years.

treatment persistence rates at two years were 55.6% (95%CI: 29.7–78.7) in the bio-naïve group, and 80.0% (30.9–97.3) in the switch group, indicating numerically higher treatment persistence rates in the latter group (S5 Appendix).

## Covariates associated with treatment persistence

Univariable analysis for treatment persistence rates in all patients with CD showed that albumin level (HR = 0.36, 95% CI: 0.15–0.90) and group (bio-failure VS switch, HR = 0.17, 95% CI: 0.038–0.80) were significantly associated with treatment persistence.

Although multivariable analysis adjusted for age and sex for treatment persistence rates in all patients with CD showed that albumin level (HR = 0.40, 95% CI: 0.16–1.01) was not significantly associated with treatment persistence, it was well consistent with the result of the

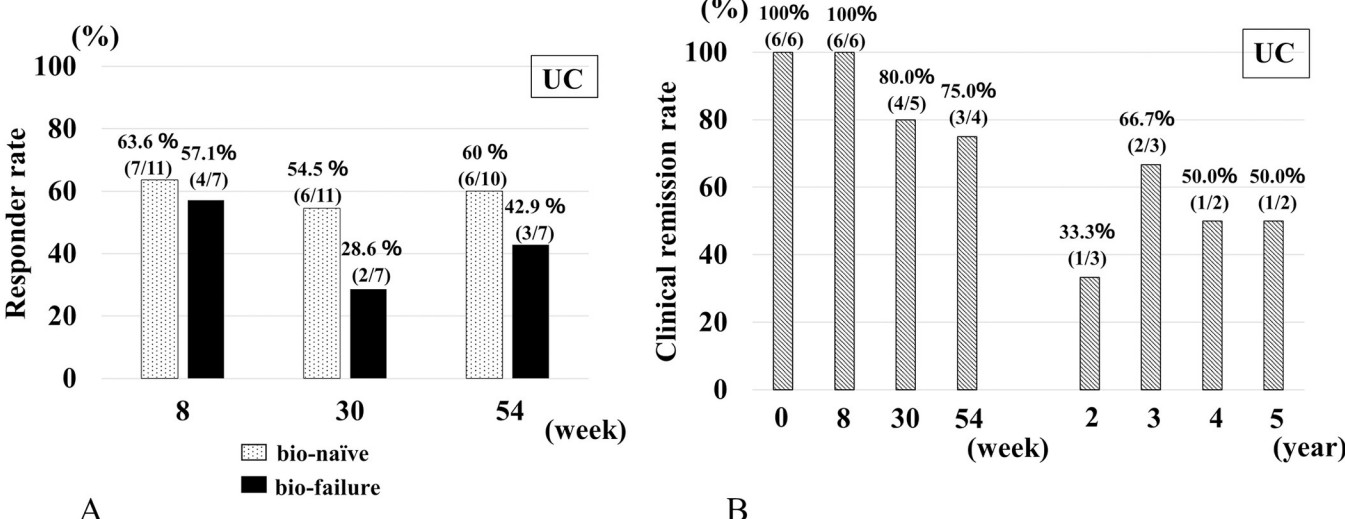

**Fig 2. Effectiveness of IFX-BS in patients with UC.** (A) Responder rate in patients with UC at baseline, 8, 30, and 54 weeks. (B) Clinical remission rate in patients with UC at baseline, 8, 30, and 54 weeks, and 2, 3, 4, and 5 years.

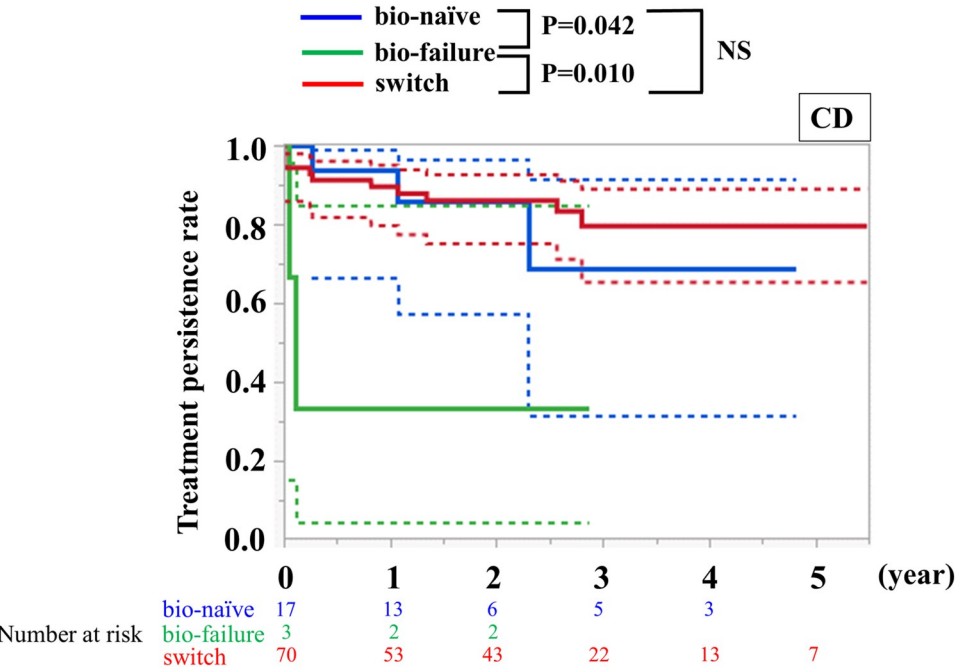

**Fig 3. Kaplan–Meier curve of treatment persistence.** Kaplan–Meier curve of treatment persistence in patients with CD. Statistical significance was analyzed using the log-rank test. 95% confidence intervals are shown in dot lines. NS: Not Significant. Treatment persistence rates at two years were 85.9% (95%CI: 57.4–96.5) in the bio-naïve group and 86.0% (75.0–92.6) in the switch group.

univariable analysis. Group (bio-failure VS switch, HR = 0.18, 95% CI: 0.035–0.96) was significantly associated with treatment persistence as in the univariable analysis (Tables 4 and 5).

## Safety

The adverse events are shown in Table 6, among which the most common of which was infusion reaction (4.2%). Three patients had drug-induced interstitial pneumonia, sarcoidosis-like

**Table 4. Univariable cox regression analysis used to identify the covariates associated with treatment persistence in CD patients.**

| Covariates | HR (95%CI) |
| --- | --- |
| Sex | 1.74 (0.63–4.81) |
| Age | 0.98 (0.94–1.02) |
| Group[a] (bio-naïve VS bio-failure) | 5.46 (0.87–34.1) |
| Group[a] (bio-naïve VS switch) | 0.90 (0.47–1.70) |
| Group[a] (bio-failure VS switch) | 0.17 (0.038–0.80) |
| Hemoglobin | 0.86 (0.66–1.12) |
| Albumin | 0.36 (0.15–0.90) |
| CRP | 1.17 (0.99–1.38) |
| Concomitant immunomodulator | 0.84 (0.31–2.31) |

All covariates were assessed at the initial administration of IFX-BS.

[a]Group indicate bio-naïve, bio-failure, and switch groups.

Sex: nominal variable; Age (year): continuous variable; Group: nominal variable; Hemoglobin (g/dL): continuous variable; Albumin (g/dL): continuous variable; CRP (mg/dL): continuous variable; concomitant immunomodulator: nominal variable

**Table 5. Multivariable analysis adjusted for age and sex used to identify the covariates associated with treatment persistence in CD patients.**

| Covariates | HR (95%CI) |
|---|---|
| Group[a] (bio-naïve VS bio-failure) | 11.4 (0.87–148.7) |
| Group[a] (bio-naïve VS switch) | 0.91 (0.49–1.68) |
| Group[a] (bio-failure VS switch) | 0.18 (0.035–0.96) |
| Hemoglobin | 0.87 (0.65–1.17) |
| Albumin | 0.40 (0.16–1.01) |
| CRP | 1.16 (0.96–1.41) |
| Concomitant immunomodulator | 0.91 (0.33–2.51) |

All covariates were assessed at the initial administration of IFX-BS.

[a]Group indicate bio-naïve, bio-failure, and switch groups.

Group: nominal variable; Hemoglobin (g/dL): continuous variable; Albumin (g/dL): continuous variable; CRP (mg/dL): continuous variable; Concomitant immunomodulator: nominal variable

granulomatous lung disease, and pulmonary tuberculosis. Five patients developed cancers (one case each of hepatocellular carcinoma, breast cancer, malignant lymphoma, and two of rectal cancer) during the observational period.

## Discussion

This is the first retrospective cohort study to investigate the long-term effectiveness and safety of IFX-BS beyond five years in Japanese patients with IBD with respect to clinical practice. We found that the effectiveness of remission induction and maintenance by IFX-BS was comparable to that by IFX originator [1, 2]. We also presented significant data regarding the treatment persistence rate of IFX-BS about two years, which for the bio-naïve group in patients with CD

**Table 6. Adverse events in patients with CD and UC treated with IFX-BS.**

| Adverse event | n (%) |
|---|---|
| Any adverse event | 18 (15.4) |
| Infusion reaction | 5 (4.2) |
| **Lung disease** | 3 |
| Drug-induced Interstitial pneumonia | 1 (0.7) |
| Drug-induced lung disease (Sarcoidosis-like granulomatous lung disease) | 1 (0.7) |
| Pulmonary tuberculosis | 1 (0.7) |
| **Malignancy** | 5 |
| Malignant lymphoma | 1 (0.7) |
| Breast cancer | 1 (0.7) |
| Hepatocellular carcinoma | 1 (0.7) |
| Rectal cancer | 2 (1.7) |
| Allergic symptoms | 2 (1.7) |
| Arthralgia | 1 (0.7) |
| Paradoxical reaction of hand | 1 (0.7) |
| Abdominal pain | 1 (0.7) |
| IgA nephropathy | 1 (0.7) |
| Diarrhea | 1 (0.7) |
| General fatigue | 1 (0.7) |
| Jaundice | 1 (0.7) |

(85.9%) was comparable to that of the switch group (86.0%). Furthermore, univariable and multivariable analyses for treatment persistence rate demonstrated that albumin level and groups (bio-naïve, bio-failure and switch) were effective factors in patients with CD.

Several reports indicated absence of difference in the effectiveness and safety between IFX-BS and IFX originators in induction remission [15–17]. Other studies have reported effectiveness and safety of switching from originators to IFX-BS [18]. Based on these clinical data, the introduction of IFX-BS has been recommended in an attempt to reduce national healthcare costs [19]. The market share of IFX-BS was 89% in the U.K [20], 76% in France [21], 85.3% in South Korea [22], and 9.7% in the U.S [23]. In Japan, the market share of IFX-BS has been gradually increasing and accounted for 25% of the total IFX in 2018 [24]. Despite the increasing use of IFX-BS worldwide, clinical data on the long-term effectiveness of IFX-BS treatment, such as the five years treatment persistence rate, is insufficient. Therefore, we investigated the effectiveness and safety of both short- and long-term administration of IFX-BS in practical clinical practice based on the Phoenix cohort data.

First, we evaluated the effectiveness of IFX-BS in patients with IBD during the induction phase. In patients with CD, the clinical response rates in the bio-naïve and bio-failure groups at 8 weeks were 81.8% (9/11) and 0% (0/3), respectively. The reason for the non-responsiveness of bio-failure patients with CD to IFX-BS largely depended on them receiving another anti-TNF-α agents treatments. On the contrary, in patients with UC, there was no significant difference in the clinical response rate between the bio-naïve and bio-failure groups. One possible reason for the bio-failure group in UC also showing a better response to IFX-BS as compared to that in CD is the difference in the prior treatment. IFX-BS can be effective in patients who switched from adalimumab because of the increase in antibody agent according to patient weight, resulting in a relative increase in treatment intensity. In addition, IFX-BS can be effective in cases of LOR with an anti-adalimumab antibody. Furthermore, in cases previously treated with vedolizumab or tofacitinib, where IFX-BS was the first anti-TNF-α agent, it can be effective.

In the maintenance phase, the treatment persistence rate of the bio-naïve group in patients with CD at two years (85.9%) was comparable to that of the switch group (86.0%). Blesl et al. reported that the two years treatment persistence rate of IFX originator for CD was 58% (95% CI: 53–63), and Keshavarzian A et al. reported it was 73.0% [25, 26]. Despite a wide range of the treatment persistence rate reported previously, the treatment persistence rate of IFX-BS in this study was not inferior to that of the originator. This implies that the switch to IFX-BS is acceptable in patients who have been in remission with anti-TNF-α agents, which primarily contributes to the pathogenesis of CD.

Interestingly, we found that the treatment persistence rate of IFX-BS in the UC switch group was higher than in the UC bio-naïve and bio-failure groups during the long-term follow-up period (two years). This data also supports that a switch to IFX-BS for UC in remission with IFX originator is acceptable. Of note, Blesl et al. also reported that treatment persistence was longer in CD than in UC [25]. These data may indicate heterogeneity of UC pathophysiology.

Univariable analysis of treatment persistence rate showed that albumin level at the initial administration of IFX-BS and groups were significantly associated with treatment persistence. No significant difference in albumin level was observed according to multivariable analysis adjusted for age and sex, but as the HR did not change a lot, it is thought to have contributed to the treatment persistence rate. Regarding the factors associated with long-term treatment persistence of anti-TNF-α agent originator, CD had significantly higher treatment persistence rates than those of UC and serum albumin level was an independent predictor of treatment persistence in CD. High dose administration of anti-TNF-α agents originator was reported as

another predictor of treatment persistence [25, 27]. Regarding the factors associated with short-term treatment persistence of IFX-BS, combination with steroid, CRP levels < 0.5 mg/dL and serum albumin levels ≥ 3.5 g/dL at baseline were independently associated with better treatment persistence in patients with CD [28]. Further studies are needed to identify the appropriate cases for IFX-BS administration.

The clinical remission rates for the switch group of CD were also analyzed using the IPCW method (S1 Table). The results showed a higher remission rate than that of Fig 1B. In this cohort, almost all patients of censored cases were able to continue IFX-BS treatment, which could be the reason for the difference between the original data and data obtained by the IPCW method. The treatment of missing scores as censored could potentially lead to bias.

This study has several limitations. First, the sample size was relatively small, especially in UC patients and the bio-failure groups of CD and UC patients. UC patients has more treatment options, which could be the reason why we had a small number of UC patients in this retrospective cohort data. Second, our data were obtained from only three facilities in Japan. A multicenter study with a larger sample size is needed to evaluate the long-term efficacy and safety of IFX-BS.

## Conclusion

In summary, the present study showed that IFX-BSs are as effective and safe as IFX originators even with long-term administration. In addition to the favorable remission induction in the bio-naïve and bio-failure groups, we demonstrated the remission maintenance and treatment persistence rates beyond two years after treatment. Moreover, we showed that the albumin level and groups were associated with better treatment persistence in patients with CD.

## Supporting information

**S1 Appendix. Change in CDAI scores in patients with CD at baseline, 8, 30, and 54 weeks.**
(TIF)

**S2 Appendix. Change in CRP levels in patients with CD at baseline, 8, 30, and 54 weeks.**
(TIF)

**S3 Appendix. Change in pMayo scores in patients with UC at baseline, 8, 30, and 54 weeks.**
(TIF)

**S4 Appendix. Change in CRP levels in patients with UC at baseline, 8, 30, and 54 weeks.**
(TIF)

**S5 Appendix. Kaplan–Meier curve of treatment persistence in patients with UC.** Statistical significance was analyzed using the log-rank test. NS: Not Significant.
(TIF)

**S1 Table. Clinical remission rates of patients with CD at 30, and 54 weeks, and 2, 3, 4, and 5 years corrected for censored data using the IPCW method.**
(TIF)

## Acknowledgments

We thank Kohei Wagatsuma, Daisuke Hirayama, Yoshihiro Yokoyama, and Tsukasa Yamakawa for acquiring the data and Yuki Hayashi for statistical analysis.

## Author Contributions

**Conceptualization:** Tomoe Kazama, Keisuke Ishigami, Hiroshi Nakase.

**Data curation:** Tomoe Kazama, Keisuke Ishigami.

**Formal analysis:** Tomoe Kazama, Keisuke Ishigami, Masanori Nojima.

**Investigation:** Tomoe Kazama, Keisuke Ishigami.

**Methodology:** Tomoe Kazama, Keisuke Ishigami.

**Project administration:** Hiroshi Nakase.

**Resources:** Tomoe Kazama, Katsuyoshi Ando, Nobuhiro Ueno, Mikihiro Fujiya, Takahiro Ito, Atsuo Maemoto, Hiroshi Nakase.

**Supervision:** Hiroshi Nakase.

**Validation:** Tomoe Kazama.

**Visualization:** Tomoe Kazama.

**Writing – original draft:** Tomoe Kazama, Keisuke Ishigami, Hiroshi Nakase.

**Writing – review & editing:** Katsuyoshi Ando, Nobuhiro Ueno, Mikihiro Fujiya, Takahiro Ito, Atsuo Maemoto, Keisuke Ishigami, Masanori Nojima, Hiroshi Nakase.

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
