## [Decision Letter · Decision Letter 0]

27 Dec 2022

PONE-D-22-30848Long-term effectiveness and safety of infliximab-biosimilar: a multicenter Phoenix retrospective cohort study

PLOS ONE

Dear Dr. Nakase,

Thank you for submitting your manuscript to PLOS ONE. After careful consideration, we feel that it has merit but does not fully meet PLOS ONE’s publication criteria as it currently stands. Therefore, we invite you to submit a revised version of the manuscript that addresses the points raised during the review process.

We look forward to receiving your revised manuscript.

Kind regards,

Shintaro Sagami

Academic Editor

PLOS ONE

Journal Requirements:

"H Nakase has received support from AbbVie, Celgene, Daiichi Sankyo, EA Pharma, Janssen, JIMRO, Kissei Pharmaceutical, Kyorin Pharmaceutical, Mitsubishi Tanabe Pharma, Mochida Pharmaceutical, Nippon Kayaku, Pfizer, Takeda, and Zeria Pharmaceutical, as well as grants for commissioned/joint research from Boehringer Ingelheim, Bristol Myers Squibb, and Pentax Medical.

Dr. Fujiya acknowledges grants from Japanese Grants-in-Aid for Scientific Research (21K07929), grants from Translational Research Network Program of Japan Agency for Medical Research and Development (B-118), non-financial support from Development and Intractable Disease Health and Labour Sciences Research Grants from the Ministry of Health, Labour and Welfare.

Dr. Fujiya reports grants, personal fees from EA Pharma Co., Ltd., grants, personal fees from AYUMI Pharmaceutical Corporation, grants, personal fees from AbbVie Inc, grants, personal fees from Otsuka Pharmaceutical Co., Ltd., grants, personal fees from ZERIA Pharmaceutical Co., Ltd., grants, personal fees from Nippon Kayaku Co., Ltd., grants, personal fees from Nobelpharma Co., Ltd., grants, personal fees from Pfizer Inc, grants, personal fees from Janssen Pharmaceutical K.K., grants, personal fees from KYORIN Pharmaceutical Co., Ltd., grants, personal fees from MOCHIDA PHARMACEUTICAL CO.,LTD., grants, personal fees from Daiichi Sankyo Company, Limited, grants, personal fees from Mitsubishi Tanabe Pharma Corporation, grants, personal fees from Takeda Pharmaceutical Company Limited, grants, personal fees from Yakult Honsha Co., Ltd., personal fees from OLYNPUS Co., Ltd., personal fees from celltrionhealthcare.jp, personal fees from Alfresa Pharma Corporation, personal fees from Mylan Inc., personal fees from Boston Scientific Corporation, personal fees from Covidien Japan, Inc., personal fees and non-financial support from FUJIFILM Corporation, grants from Fuji Chemical Industries Co., Ltd., grants from JIMRO Co., Ltd., grants from Kamui Pharma. Inc.,

K Ando has received lecture fees from Nippon Kayaku Co. Ltd., Janssen Pharmaceutical K.K., Mitsubishi Tanabe Pharma Corporation, , AYUMI Pharmaceutical Corporation, Pfizer Inc., Takeda Pharmaceutical Co. Ltd., AbbVie GK., JIMRO Co. Ltd., EA Pharma Co. Ltd., Mochida Pharmaceutical Co. Ltd., Kyorin Pharmaceutical Co. Ltd., ZERIA Pharmaceutical Co. Ltd., Aspen Japan K.K., Sandoz K.K. and research grant from Pfizer Inc.

N Ueno has received personal fees from AbbVie Inc, AYUMI Pharmaceutical Corporation, EA Pharma Co. Ltd., Janssen Pharmaceutical K.K., JIMRO Co. Ltd., Kyorin Pharmaceutical Co. Ltd., Mitsubishi Tanabe Pharma Corporation, Mochida Pharmaceutical Co. Ltd., Nippon Kayaku Co. Ltd., and Takeda Pharmaceutical Co. Ltd.

A Maemoto and T Ito have received support from Takeda Pharmaceutical Company, Eli Lilly Japan K.K., Janssen Pharmaceutical K.K., Gilead Sciences, Inc., Nippon Boehringer Ingelheim Co., Ltd., AbbVie GK, Pfizer R&D Japan G.K., EA Pharma Co., Ltd., KakenPharmaceutical Co., Ltd., Mochida Pharmaceutical Co., Ltd., Kissei Pharmaceutidal Co., Ltd., Mitsubishi Tanabe Pharma Corporation."

Additional Editor Comments:

-As the Reviewer #1 mentioned, the treatment of missing data is crucial. Please describe in detail in the Methods. Also consider why they are missing, such as whether they have changed treatment or stopped coming to the hospital.

-The persistence rate should vary considerably depending on whether it is Naive, Switch, or Failure. It is insufficient that this variable is not included in the multiple regression analysis. The information of Biologics Failure might be worse predictor than combo therapy.

-Have you gathered any information on whether the patients used immunomodulator before the baseline or immediately after starting treatment? If you have information when immunomodulator was concomitantly used, please specify. Does the effectiveness change depending on the timing of concomitant use?

-This is likely an unstable model because multivariate analysis was performed despite the small number of no-persistence cases. Since it changes depending on how the missing values were handled, could you also present as supplementary data whether each factor in the Cox proportional hazards was a significant factor? If you do not have the required number of cases, I would be willing to do just a univariate analysis.

Reviewers' comments:

Reviewer's Responses to Questions

**Comments to the Author**

1. Is the manuscript technically sound, and do the data support the conclusions?

Reviewer #1: Partly

Reviewer #2: Yes

2. Has the statistical analysis been performed appropriately and rigorously? 

Reviewer #1: Yes

Reviewer #2: Yes

3. Have the authors made all data underlying the findings in their manuscript fully available?

Reviewer #1: Yes

Reviewer #2: Yes

4. Is the manuscript presented in an intelligible fashion and written in standard English?

Reviewer #1: Yes

Reviewer #2: Yes

5. Review Comments to the Author

Reviewer #1: This study by Kazama et al provides new evidence about the effectiveness of infliximab biosimilars in IBD, in various contexts : at initiation, at medical switch and at non medical switch. Despite the relatively small number of patients included, the authors show different persistence in between groups and highlight predicting factors for persistence under treatment.

Abstract

1. P2 l21 The author mentions « biosimilar » in singular : Japan has not approved more than one biosimilar product for infliximab ? CTP13 ? SB2 ?

2. P2 It would be good to add the dates of commercialization of infliximab originator and biosimilar products

3. P2 l38 Typo : findngs => findings

4. P2 It would have been interesting to compare the effectiveness and adverse events rates in patients taking the originator product. The authors could at least cite the persistence rates known for the originator product

Introduction

1. P4 l62 : IFX-BSs are not approved for psoriasis ?

2. Please comment on the regulatory process for biosimilar approval and the specific process in Japan

Material and Methods

1. P4 l81 : as the objective is to assess the long term persistence of patients, why didn’t the authors stopped the inclusionsat least 1 or 2 years before the end of study ?

2. P5 l90 : what is the difference between the biofailure group that took IFX originator or adalimumab and the switch group ? Could the switch other anti-TNF alpha to IFX BS be non medical and comparable to the switch originator IFX to IFX BS ?

3. P5 L96 : How did you deal with lost of follow-up ? with censoring ?

4. P6 l111 : define the CR here (it is written on line 113)

5. P6 L117 : “We also evaluated the clinical response in the bio-naïve and bio-failure groups” this sentence is unclear : clinical response is not evaluated in all the groups ? the beginning of the paragraph seemed to say so…

6. P6 L117 : can you please define the events of interest taken into account for safety ?

7. P6 L120 : Please define CRP

8. P6 L128 : as the authors are modelling persistence time, wouldn’t it have been appropriate to model the risk of non persistence with a Cox regression or a Poisson regression ? as the authors are dealing with small sample sizes, wouldn’t it have been appropriate to use Bayesian methods ?

9. Please cite the “Kaplan Meier” methodology for persistence modelling.

10. P7 l129 : please clarify the methodology for the logistic regression modelling the factors in favor of persistence : do you compare “at least 1 year persistent patients” to “less than 1 year persistent patients” ? Did you carry out sensitivity analyses on 2/3/4/5 years persistent patients ? Did you try to look at low / medium / high persistent patients ?

Results

1. Table 1 : What does “range” mean ? interquartile range ? min-max range ?

2. Table 1 : back to the methods “The switch group included patients who received IFX originator or other anti-TNF-α agents in remission at baseline and switched to IFX-BS” : finally, the switch group is exclusively composed of IFX originator => IFX BS switchers right ?

3. Table 1 (and other results) : as your sample size is quite small, please consider using only 2 significant digits in the percentages calculations

4. Table 1 : please add the year of inclusion

5. Table 1 is not organized like state of the art Table 1, and thus quite hard to read.

6. Figures : CDAI score, CRP, and pMayo score can be put in supplementary material

7. Figure legends : please specify in the legend the group of patients which is considered. CR was evaluated for switch group only ? Please clarify in the Methods section.

8. Is it relevant to show the results of CD and UC separately ? The number of UC patients is very small and thus at risk of sampling fluctuation.

9. Kaplan meier curves : please show the number of patients still at risk throughout the follow-up period. Please add the confidence intervals / bands. This seems to be important figures to highlight more.

10. P12 l199 : please show the confidence intervals for persistence rates

11. Table 2/3 : please consider using a model with UC and CD patients together, adjusting on the pathology

12. Table 2/3 : how did you categorized the variables hemoglobin, albumin, CRP ? Is it based on clinical thresholds ?

13. Table 2/3 : please show the effectives in each category and the shares of persistent vs non persistent

14. Please show the number of person years, mean and median follow up durations, for each group and each disease

15. P13 L220 : no conclusion can be drawn on UC patients

16. P16 l240 : please show the incidence of adverse events relatively to the follow up duration

Discussion

1. P17 l252 : as the authors did not compare head to head biosimilar and originator, the statement “We found that the effectiveness of remission induction and maintenance by IFX BS was comparable to that by IFX originator” is misleading and exaggerated

2. P17 L255 : the CRP was not strictly significantly associated with persistence

3. P17 l261 : the figure for France in incorrect

https://www.nature.com/articles/s41598-022-24050-7 ; https://www.sciencedirect.com/science/article/abs/pii/S0040595721002596?via%3Dihub

4. P18 l 275 : “One possible reason for the bio-failure group in UC also showing a better response to IFX-BS as compared to that in CD is the difference in the prior treatment. IFX-BS can be effective in patients who switched from adalimumab because of the increase in antibody agent according to patient weight, resulting in a relative increase in treatment intensity. In addition, IFX-BS can be effective in cases of LOR with an anti adalimumab antibody. Furthermore, in cases previously treated with vedolizumab or tofacitinib, IFX-BS was the first anti-TNF-α agent and can be effective” is there some literature on this topic ?

5. P18 l282: I am not very comfortable with your statement, as bio naïve patients are by construction more subject to treatment failure than switching patients that are stabilized by the originator treatment… Once again, you did not compare switchers with non switchers, so you cannot conclude unfortunately.

6. P19 l305 : “There can have been a bias in the cases” please clarify

7. P19 l305 “switch group was expected to have included more patients with CD at the time of IFX administration” please clarify

8. Limitation : study restricted to 3 japanese centers

Reviewer #2: It is my pleasure to review the manuscript entitled "Long-term effectiveness and safety of infliximab-biosimilar: a multicenter Phoenix retrospective cohort study".

However, I have several concerns that need to be addressed before considering publication.

1) The authors write that "Univariate analysis for persistence rate demonstrated that concomitant use of immunomodulator (OR=3.90, 95% CI: 0.77-19.70, P=0.09) and low CRP levels during initial administration of IFX-BS (OR=3.97, 95% CI: 1.04-15.14, P=0.04) were associated with persistence in patients with CD." on line 214-217, page 13. I think "concomitant use of immunomodulator" is not significantly associated with persistence in patients with CD.

2) The authors show the responder rate in patients with UC at baseline, 8, 30, and 54 weeks in Figure 3A. In bio-naive group, the denominator at 8 weeks is 11. And the denominator at 30 weeks is 14. It is unnatural that denominator increases over time. Please check the study data.

3) I think it might be better to mention the duration of administration of IFX-BS and observation period of each group (All CD patients, bio-naïve group with CD, bio-failure group with CD, switch group with CD, All UC patients, bio-naïve group with UC, bio-failure group with UC, switch group with UC).

6. PLOS authors have the option to publish the peer review history of their article (what does this mean?). If published, this will include your full peer review and any attached files.

Reviewer #1: **Yes: **Hugo JOURDAIN

Reviewer #2: **Yes: **Sakuma Takahashi

---

## [Author Response · Author response to Decision Letter 0]

10 Feb 2023

Reviewer＃1 We agree with you and have incorporated this suggestion throughout our paper. Thank you very much.

Reviewer#2 Thank you very much for your appropriate comments. I have incorporated your suggention. 

Editor You have raised an important question. Thank you for your suggesttion.

---

## [Decision Letter · Decision Letter 1]

14 Mar 2023

PONE-D-22-30848R1Long-term effectiveness and safety of infliximab-biosimilar: 

a multicenter Phoenix retrospective cohort studyPLOS ONE

Dear Dr. Nakase,

Thank you for submitting your manuscript to PLOS ONE. After careful consideration, we feel that it has merit but does not fully meet PLOS ONE’s publication criteria as it currently stands. Therefore, we invite you to submit a revised version of the manuscript that addresses the points raised during the review process.

ACADEMIC EDITOR: The authors have generally responded accurately to the reviewers' comments, but some additional corrections and explanations are needed. Please make additional corrections to the reviewers' comments.==============================

We look forward to receiving your revised manuscript.

Kind regards,

Shintaro Sagami

Academic Editor

PLOS ONE

Journal Requirements:

Additional Editor Comments (if provided):

-While the authors assume that missing clinical scores are censored, it is generally more appropriate to perform the multiple-imputation method. Furthermore, it's essential to acknowledge the potential bias that may arise from treating a missing score as censored.

-The Kaplan-Meier method usually uses the log-rant test instead of the Wilcoxon test because the log-rank test is generally more powerful and appropriate for detecting differences between groups with respect to time-to-event outcomes. Could you explain why you choose the Wilcoxon test?

Reviewers' comments:

Reviewer's Responses to Questions

**Comments to the Author**

1. If the authors have adequately addressed your comments raised in a previous round of review and you feel that this manuscript is now acceptable for publication, you may indicate that here to bypass the “Comments to the Author” section, enter your conflict of interest statement in the “Confidential to Editor” section, and submit your "Accept" recommendation.

Reviewer #1: All comments have been addressed

Reviewer #2: All comments have been addressed

2. Is the manuscript technically sound, and do the data support the conclusions?

Reviewer #1: Partly

Reviewer #2: Yes

3. Has the statistical analysis been performed appropriately and rigorously? 

Reviewer #1: I Don't Know

Reviewer #2: Yes

4. Have the authors made all data underlying the findings in their manuscript fully available?

Reviewer #1: Yes

Reviewer #2: Yes

5. Is the manuscript presented in an intelligible fashion and written in standard English?

Reviewer #1: Yes

Reviewer #2: Yes

6. Review Comments to the Author

Reviewer #1: 1) Change “multivariate” for “multivariable”

2) Please state the variables (and units / categories) included in the multivariable cox regressions models at the bottom of table 2

3) Change “predictors” for “covariates associated with”

4) For the Bayesian method, I was simply suggesting to use Bayesian methodology to reduce : introducing a prior distribution for the HR / OR, calculating the posterior distribution, getting credibility intervals rather than 95% CI that are not relevant for small populations… I find the BMA quite confusing, maybe not explained enough.

5) Figures are displayed in low quality ?

6) Table 1 : split into 2/3 tables, not to mix the group data (naïve/failure/switch) and the complete data

7) When displaying 95% CI, do not show p values

8) In patients with CD, persistence rates at 2, 3, and 5 years were 68.8% (95%CI: 31.6-91.3), 68.8% (31.6-91.3), 68.8% (4.8 years, 31.6-91.3) in the bio-naïve group, and 77.6% (66.1-86.1), 66.0% (51.5-78.1), 66.0% (51.5-78.1) => weird to show 3 times the same persistence rates => show only the 2-year persistence (not enough patients to state the 5 years persistence)

9) “In patients with UC, the persistence rates at 2, 3, and 5 years were 43.1% (95%CI: 19.0-71.0), 43.1% (11.4-71.0), 43.1% (11.4-71.0) in the bio-naïve group, and 66.7% (26.8-91.6), 66.7% (26.8-91.6), and 66.7% (26.8-91.6) in the switch group,” => weird to show 3 times the same persistence rates => show only the 2-year persistence (not enough patients to state the 5 years persistence)

Reviewer #2: It is my pleasure to review the manuscript entitled "Long-term effectiveness and safety of infliximab-biosimilar: a multicenter Phoenix retrospective cohort study" again. The manuscript c

However, I have several concerns that need to be addressed before considering

publication.

1 Some graphs indicate the result of CD patients and the others indicate the result of UC patients.

I found it a little confusing. How about add simple explanations to the figures?

2 Boxplots in appendix are difficult to understand. What do the white circles mean?

3 Typo : On the contary => On the contrary (page 8, lines 154)

4 Typo : showen => shown (page 14, lines 213)

5 Typo : Kissei Pharmaceutidal Co., Ltd. => Kissei Pharmaceutical Co., Ltd. (Competing Interests)

7. PLOS authors have the option to publish the peer review history of their article (what does this mean?). If published, this will include your full peer review and any attached files.

Reviewer #1: **Yes: **Hugo JOURDAIN

Reviewer #2: **Yes: **Sakuma TAKAHASHI

---

## [Author Response · Author response to Decision Letter 1]

6 Jun 2023

Thank you to the editors and reviewers for their helpful suggestions. We have answered your questions. Please check it.

---

## [Decision Letter · Decision Letter 2]

26 Jun 2023

Long-term effectiveness and safety of infliximab-biosimilar: a multicenter Phoenix retrospective cohort study

PONE-D-22-30848R2

Dear Dr. Nakase,

We’re pleased to inform you that your manuscript has been judged scientifically suitable for publication and will be formally accepted for publication once it meets all outstanding technical requirements.

Kind regards,

Shintaro Sagami

Academic Editor

PLOS ONE

Additional Editor Comments (optional):

Dear Dr. Nakase,

Thank you for submitting your manuscript to our journal for review. We have completed the initial round of reviews and are pleased with the progress of your work. There are some minor revisions that need to be addressed in order to proceed.

One of our reviewers (Reviewer 2) has made a specific suggestion about the S1 Table in your manuscript. Please revise this table as per the recommendations from Reviewer 2. The revisions are considered essential for the final acceptance of your research paper. 

Upon receipt of the revised manuscript, we will expedite the review process and aim for prompt publication.

We look forward to receiving your revisions.

Best regards,

Shintaro Sagami

Reviewers' comments:

Reviewer's Responses to Questions

**Comments to the Author**

1. If the authors have adequately addressed your comments raised in a previous round of review and you feel that this manuscript is now acceptable for publication, you may indicate that here to bypass the “Comments to the Author” section, enter your conflict of interest statement in the “Confidential to Editor” section, and submit your "Accept" recommendation.

Reviewer #1: All comments have been addressed

Reviewer #2: All comments have been addressed

2. Is the manuscript technically sound, and do the data support the conclusions?

Reviewer #1: Yes

Reviewer #2: Yes

3. Has the statistical analysis been performed appropriately and rigorously? 

Reviewer #1: Yes

Reviewer #2: I Don't Know

4. Have the authors made all data underlying the findings in their manuscript fully available?

Reviewer #1: Yes

Reviewer #2: Yes

5. Is the manuscript presented in an intelligible fashion and written in standard English?

Reviewer #1: Yes

Reviewer #2: Yes

6. Review Comments to the Author

Reviewer #1: Can you please just check the log-linearity of the continuous variables you put in the Cox model ? In case they would not be log-linear, consider using splines / fractional polynomes / categorical varibales. Thanks !

Reviewer #2: It is my pleasure to review the manuscript entitled "Long-term effectiveness and safety of infliximab-biosimilar: a multicenter Phoenix retrospective cohort study" again. The manuscript has been revised well. However, I have several concerns that need to be addressed before considering publication.

1) In S1 Table, the authors have better change from ”30week" to ”30 weeks".

2) In S1 Table, the authors have better change from ”54week" to ”54 weeks".

3) In S1 Table, the authors have better change from ”2-year" to ”2 years".

4) In S1 Table, the authors have better change from ”3-year" to ”3 years".

5) In S1 Table, the authors have better change from ”4-year" to ”4 years".

6) In S1 Table, the authors have better change from ”5-year" to ”5 years".

7) Typo : serum albumin levels ≥3.5 mg/dL => serum albumin levels ≥3.5 g/dL (page 21, lines 343).

8) There are two types (mg/L and mg/dL) of measurement unit of CRP. Please be consistent with the terms you use.

7. PLOS authors have the option to publish the peer review history of their article (what does this mean?). If published, this will include your full peer review and any attached files.

Reviewer #1: **Yes: **Hugo JOURDAIN

Reviewer #2: **Yes: **Sakuma TAKAHASHI

---

## [Editor Report · Acceptance letter]

4 Sep 2023

PONE-D-22-30848R2 

Long-term effectiveness and safety of
infliximab-biosimilar:
a multicenter Phoenix retrospective cohort study 

Dear Dr. Nakase:

I'm pleased to inform you that your manuscript has been deemed suitable for publication in PLOS ONE. Congratulations! Your manuscript is now with our production department. 

Kind regards, 

on behalf of

Dr. Shintaro Sagami 

Academic Editor

PLOS ONE